



# Multidecadal sea level rise hiatus in the tropical Atlantic margin off northwest Africa

Hamed D. Ibrahim[1,2,3] and Yunfang Sun[4]

[1]University of Toronto, Department of Civil & Mineral Engineering, Toronto, ON, Canada
[2]University of Toronto, School of the Environment, Toronto, ON, Canada
[3]University of Toronto, Department of Physics, Toronto, ON, Canada
[4]University of South Florida, College of Marine Science, St Petersburg, FL, USA

**Correspondence:** Hamed D. Ibrahim (hameddibrahim@gmail.com, hamed.ibrahim@utoronto.ca) and Yunfang Sun
(yunfangsun@gmail.com)

**Abstract.** Satellite and reanalysis data sets are used to find and explain the drivers of a multidecadal sea level rise hiatus in
the tropical North Atlantic margin off northwest Africa. The study domain sea level was rising as far back as 1986, and the
hiatus began around 2010 and stopped in 2019. Mean sea level anomaly during a rising period (1996–2004) was compared to
the hiatus period (2010–2018). Results show that domain-wide seawater expansion owing to shifts in density structure (steric

shifts) contributed 74% of the multidecadal sea level shift and increase in mass contributed 22%. There are, however, regional
differences in the steric and mass shift patterns. In the northern subdomain, the steric shift is dominated by salinity-driven
(halosteric) expansion, whereas in the southern subdomain the steric shift is dominated by temperature-driven (thermosteric)
expansion. Stronger anticyclonic circulation shift in the Guinea Dome, a permanent upwelling region where isotherms are
displaced upwards, and the necessary adjustment of horizontal flow toward this anticyclonic sink, enable accumulation of

low-salinity water in the northern subdomain and precipitation in the southern subdomain. The low-salinity water influx to
the northern subdomain is linked to a shift in the southward-flowing Canary Current. This current was freshened by subpolar
North Atlantic waters that reached the northwest African coast via two pathways: an open ocean path that is consistent with
the Azores current, and a Western Europe coastal ocean path. These results highlight a multidecadal linkage between subpolar
salinity anomalies and tropical sea level anomalies in the North Atlantic, with a transit period of about 5.5 years.

## 1 Introduction

It is useful to specify the physical linkages between climatic events occurring in ocean margins and elsewhere in the ocean.
Understanding the operating phenomena that accomplish these long-term linkages provides predictive power to anticipate
adverse coastal change following events that have already occurred elsewhere in the ocean. More than 40% of the world's
population reside within 150 km of the coast (Reimann et al., 2023; United Nations, 2018) and seafood from coastal marine

ecosystems supplies about 15% of the protein consumed by this population (Sumaila et al., 2011). Another important reason
for characterizing events in ocean margins is that fluctuations in coastal seawater properties express shifts in the heat and





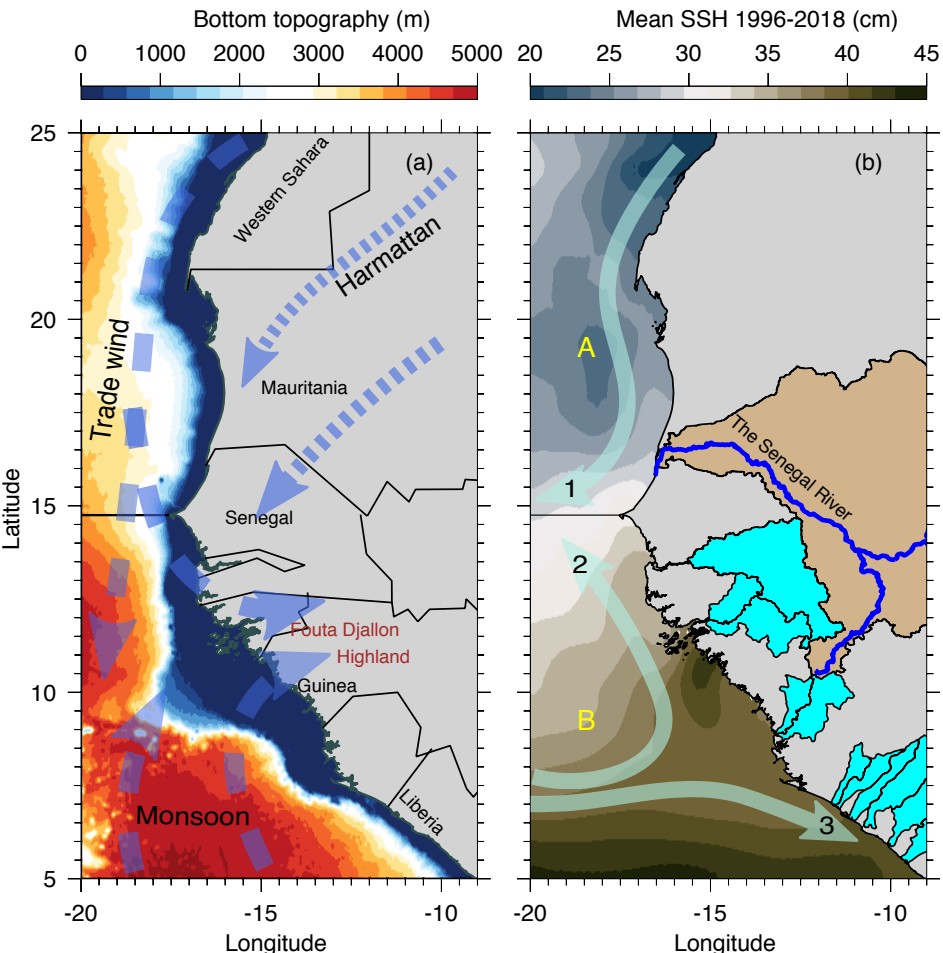

**Figure 1.** Characteristics of the eastern tropical Atlantic margin off northwest Africa (domain). (**a**) Bottom topography (GEBCO, 2021), and the mean annual spatial pattern of the three dominant atmosphere wind systems in the region (indicated in blue dash lines): trade wind, Harmattan wind, and monsoon wind. (**b**) Multi-year (1996–2018) mean annual sea-surface height ($SSH$) (C3S Climate Data Store, 2018), and the three dominant ocean current systems in the region (indicated in solid cyan lines): 1) the Canary Current, and the 2) north and 3) south branches of the bifurcated North Equatorial Countercurrent, respectively; capital letters A and B denote the two analysis subdomains identified from EOF analysis (section 2.3). The Canary Current traverses subdomain A (14.75°N–25°N, 9°W–20°W.), and the north branch of the bifurcated North Equatorial Countercurrent traverses subdomain B (5°N–14.75°N, 9°W–20°W) (Ibrahim and Sun, 2022). The regions colored in tan and cyan over land are the catchments of the rivers that discharge into subdomains A and B, respectively. See section 2.2 for more details.

water fluxes that maintain the prevailing regional climate. Analyzing satellite measurements of these fluctuations thus offers an approach to elucidate mechanisms of coastal climate change.



Sea level fluctuations in the eastern tropical Atlantic margin off northwest Africa (hereafter 'domain,' Fig. 1) is important

for climate change investigations because it reflects changes in several elements of the global ocean and atmosphere circulation (Stramma et al., 2005). These elements include the northeasterly trade wind and its continental branch, the northeasterly Harmattan wind; the southeasterly Monsoon; the south-flowing Canary Current that supplies feed-water to the North Equatorial Current; the north and south branches of the North Equatorial Countercurrent that bifurcates near the African coast (Ibrahim and Sun, 2022); and the so-called 'Guinea Dome,' a region off the African coast with permanent upward flux of cool water

from beneath that causes doming (i.e., upward displacement) of isotherms (Fig. 10). Moreover, because of persistent upwelling of nutrient-rich waters in large subregions of this domain, it is one of the three most productive marine ecosystems in the global ocean (Chavez, 2012). The complex interaction of these domain features promotes considerable interannual variability (Ibrahim and Sun, 2022) as shown in the deseasonalized time series of ocean and atmosphere quantities in the domain (Fig. 2 and Fig. 4).

The domain sea-surface temperature shifted upward in 1995 (Ibrahim and Sun, 2022). However, changepoint analysis (Lanzante, 1996) of the domain satellite altimetry measurement (Pujol et al., 2016; C3S Climate Data Store, 2018) shows a hiatus in sea level rise between two change points in 2010 and in 2019 (Fig. 2[a]). Since the 2019 change point, the domain sea level has started rising again (not shown). The satellite record of sea level anomaly ($SLA$) is relatively short, thus, to determine the trend pattern prior to the start of satellite altimetry in 1993, we used the European Centre for Medium-Range Weather Forecasts

(ECMWF) Ocean Reanalysis System 5 (hereafter 'ORAS5', see section 2.2) to also calculate the $SLA$ trend (Zuo et al., 2019). ORAS5 $SLA$ ($SLA_{\mathrm{ORAS5}}$) and satellite altimeter $SLA$ trend patterns are consistent during the overlap period (1993–2018), and $SLA_{\mathrm{ORAS5}}$ shows that the sea level there was rising as far back as 1986 (Fig. 2[a]).

Here our aim is to find and specify the drivers of the shift in observed sea level from a rising state to a hiatus state. To do this we compared the drivers of sea level shift during 1996–2004 (period one) when the sea level was rising and during 2010–2018

(period two) when the sea level rise paused. We chose period one because it overlaps with the satellite altimetry era and in order to avoid the transition period when sea level fluctuations can be comparatively unstable owing to large variations in heat and water fluxes (Ibrahim et al., 2020). We chose period two in order to assess the sea level rise hiatus. The explanation of the observed $SLA$ trend must account for the steric, mass and atmospheric pressure shifts between period one and two, since these are the three factors that control $SLA$ fluctuations (section 2.1). Moreover, because the atmospheric pressure shift is

comparatively small in general, we anticipate that steric and mass shifts play the key roles in this sea level rise hiatus.

In order to achieve this aim, we first performed an empirical orthogonal function analysis of the domain $SLA$, which revealed two subregions (denoted 'subdomain A' and 'subdomain B' in Figure 1[b]]) with differing horizontal pattern of $SLA$ variability (section 2.3). This is followed by separating the multi-year mean $SLA$ shift in each subdomain into its constituent steric, mass, atmospheric pressure shifts, thus identifying the hiatus drivers in each subdomain (section 3). Two key aspects

of the dynamical chain of events resulting in this hiatus are discussed in section 4 and section 5, respectively. Our concluding remarks are in section 6.





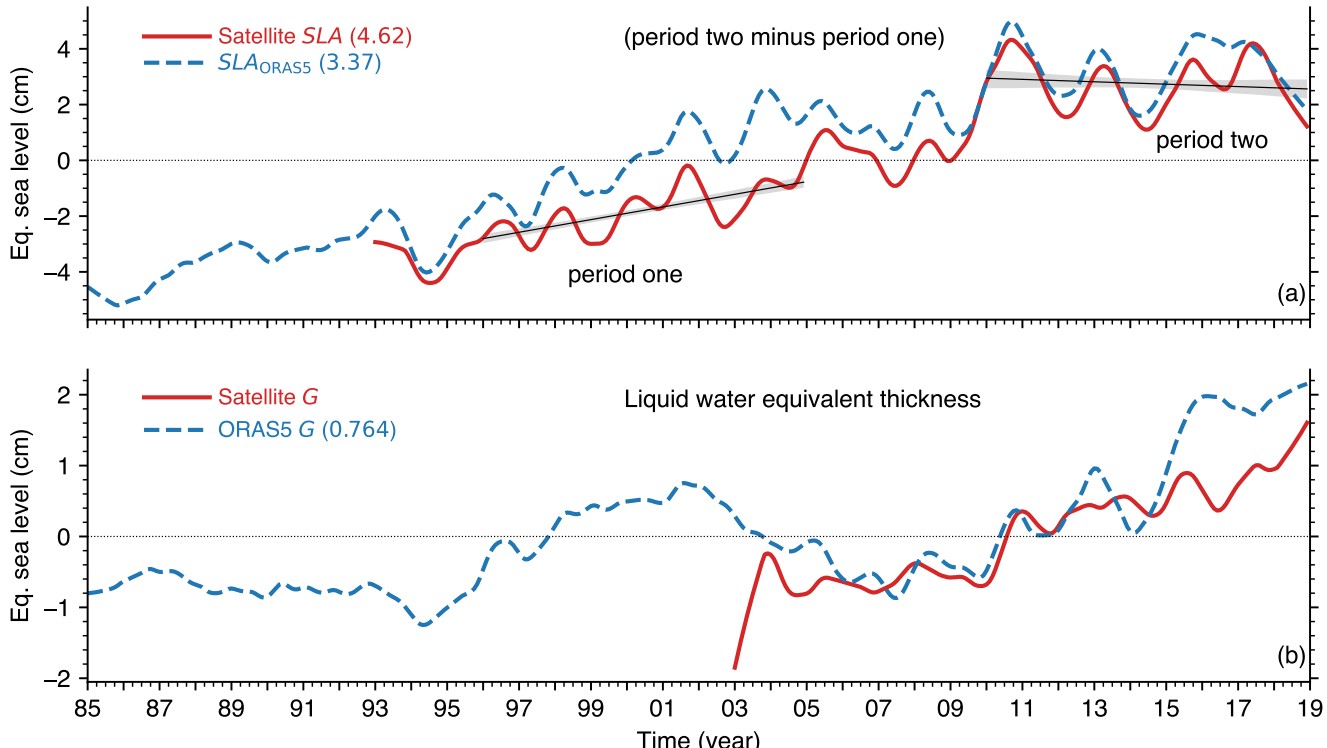

**Figure 2.** Area-average monthly time series (annual cycle removed) of domain characteristics and multi-year mean shifts (i.e., period two minus period one) in parenthesis. (**a**) Satellite altimetry sea level anomaly ($SLA$) (red solid line) with regression line (black solid line) and 95% confidence interval for each period; and ORAS5 $SLA$ ($SLA_{\mathrm{ORAS5}}$) (blue dash line); for comparison, period one and two mean difference of satellite altimetry global average sea level is 4.02 cm. (**b**) GRACE satellite (see section 2.1) ocean mass change ($G$) (red solid line) and calculated ORAS5 $G$ (blue dash line) (see section 2.2). GRACE $G$ missing months (about 10% of the data set) have been infilled with the climatology of the missing month. A 13-month low-pass (moving average) filter is applied to all the time series.

## 2 Methods of analysis

To understand and characterize the measured domain $SLA$ pattern (Fig. 2[a]), it is necessary to analyze the causes of change in the vertical displacement of the free ocean surface at each instant. The dominant causes have been known long ago and illustrated by several authors (Pattullo et al., 1955; Gill and Niller, 1973; Pinardi et al., 2014; Fukumori and Wang, 2013; Ibrahim and Sun, 2020). However, owing to the diversity of terminology in the sea level literature (Gregory et al., 2019), we briefly summarize these causes (section 2.1) to facilitate interpretation of our results. This is followed by a description of the data sets that we used to calculate the shifts in these causes (section 2.2) and the results of EOF analysis that reveal two subdomains with differing pattern of $SLA$ variability (section 2.3). Note that in section 2.1 we refer to $SLA_{\mathrm{ORAS5}}$ and ORAS5 $G$, which are both calculated using the ORAS5 reanalysis data sets that are described in section 2.2.





## 2.1 The causes of sea level fluctuations

The basic formulation of the physics of sea level fluctuation given by Gill and Niller (1973) is followed closely here and is thus referred to for completeness. Introducing $\eta\,(\lambda, \phi)$ for the free ocean surface, where $\lambda$ and $\phi$ are longitude and latitude (positive northward and eastward), respectively; $-H$ (m) for the ocean bathymetry (i.e., depth from the mean ocean surface); $\rho\,(\mathrm{kgm^{-3}})$

for seawater density; $p_{\mathrm{b}}\,(\mathrm{kgm^{-1}s^{-2}})$ for pressure at the ocean bathymetry (i.e., ocean bottom pressure); $p_{\mathrm{a}}\,(\mathrm{kgm^{-1}s^{-2}})$ for atmospheric pressure at the ocean surface, and $g\,(\mathrm{ms^{-2}})$ for the constant acceleration due to gravity; then based on the hydrostatic relation the variation of the free ocean surface from its time-mean, $\eta'$ (m), can be approximated by (Gill and Niller, 1973)

$$\underbrace{\eta'}_{SLA} = \underbrace{\left(-\frac{1}{\rho_{\mathrm{o}}}\int_{-\mathrm{H}}^{0}\rho'\mathrm{d}z\right)}_{\underbrace{Z_{\alpha}}} + \underbrace{\frac{p_{\mathrm{b}}'}{g\rho_{\mathrm{o}}}}_{G} + \underbrace{\left(\frac{-p_{\mathrm{a}}'}{g\rho_{\mathrm{o}}}\right)}_{\zeta_{\mathrm{a}}} \tag{1}$$

$$\underbrace{\phantom{\left(-\frac{1}{\rho_{\mathrm{o}}}\int_{-\mathrm{H}}^{0}\rho'\mathrm{d}z\right) + \frac{p_{\mathrm{b}}'}{g\rho_{\mathrm{o}}}}}_{SLA_{\mathrm{ORAS5}}}$$

where $\rho'$, $p_{\mathrm{a}}'$, $p_{\mathrm{b}}'$ are the variations of $\rho$, $p_{\mathrm{a}}$ and $p_{\mathrm{b}}$ from their time-mean, respectively, and $\rho_{\mathrm{o}}$ is a representative density (a

constant). The mean sea level is defined as the geopotential surface $z = 0$ where the time average fluctuation of $\eta$ is equal to zero (Apel, 1987). Eq. (1) states that $\eta'$, the sea level anomaly from its time-mean, hereafter $SLA$ (cm), is caused by: (1) shifts in density structure resulting in expansion/contraction of the seawater column, i.e., steric fluctuation, hereafter $Z_{\alpha}$ (cm); (2) variations in freshwater and salt mass (which determines weight and therefore pressure at the ocean bottom) inside the seawater column below the ocean surface, hereafter $G$ (cm); and (3) variation in atmospheric pressure at the ocean surface, hereafter

$\zeta_{\mathrm{a}}$ (cm). Note that in the first and third term on the right-hand side of Eq. (1), a negative variation (i.e., $\rho' < 0$, or $p_a' < 0$) implies an increase in $SLA$ and vice versa. As in Gill and Niller (1973), our focus is on absolute sea level shift and we do not consider vertical land displacement (i.e., relative sea level shift). The domain relative sea level shift needs to be calibrated with co-located long-term tidal guage records and GPS based altimeter observations (Wöppelmann and Marcos, 2016; Klos et al., 2019).

For consistency, because satellite $SLA$ trend pattern and ocean reanalysis $SLA_{\mathrm{ORAS5}}$ trend pattern are in agreement (Fig. 2[a]), we used $SLA_{\mathrm{ORAS5}}$ to evaluate Eq. (1). However, satellite $SLA$ (Fig. 2[a]) is corrected for $\zeta_{\mathrm{a}}$ (i.e., barometric correction), but $SLA_{\mathrm{ORAS5}}$ does not include atmospheric pressure forcing (Zuo et al., 2019), i.e., $SLA_{\mathrm{ORAS5}}$ is the sum of $Z_{\alpha}$ and $G$ as shown in Eq. (1). Since our aim is to specify the contribution of the three causes on the right side of Eq. (1) between period one and two, we therefore estimate and add $\zeta_{\mathrm{a}}$ to $SLA_{\mathrm{ORAS5}}$. Hence, hereafter, $SLA$ refers to $SLA_{\mathrm{ORAS5}}$ plus $\zeta_{\mathrm{a}}$.

Compared to $Z_{\alpha}$ and $G$, the contribution of $\zeta_{\mathrm{a}}$ to $SLA$ shifts is relatively small, so we use the approximation suggested by Gill and Niller (1973), $p_a$ shift of 1 mbar corresponds to $\eta'$ shift of 1 cm, which is consistent with direct calculations using term three on the right side of Eq. (1) and taking $\rho_o = 1025\ \mathrm{kgm^{-3}}$.

ORAS5 is a Boussinesq model, meaning that it conserves volume but not mass. Greatbatch (1994) showed that requiring the conservation of mass in Boussinesq ocean models introduces two new terms in Eq. (1): the first term is weak and can

be neglected, while the second term corresponds to the global inverse barometer effect, i.e., a spatially uniform net rate of





expansion/contraction of the global sea level. Therefore, we calculate and include this second component, hereafter referred to as the 'Boussinesq correction ($\varepsilon$),' in our analysis, i.e., $\varepsilon$ is added as the fourth term on the right-hand side of Eq. (1). However, for the sake of simplicity of description and because $\varepsilon$ is small (less than 1% of the $SLA$ shift, see Table A1), we do not write it in Eq. (1).

We used two methods to estimate $Z_\alpha$. In method one we used ORAS5 reanalysis temperature and salinity to calculate $G$ (ORAS5 $G$, see details in the next section); we then used ORAS5 $G$ and $SLA_{\mathrm{ORAS5}}$, together with Eq. (1), to obtain $Z_\alpha$ as a residual. Method one may be thought of as equivalent to taking the difference between two satellite measurements (i.e., subtracting GRACE measurement from altimetry measurement). In method two we directly calculate $Z_\alpha$, as well as its temperature-driven component (thermosteric change, hereafter $Z_\mathrm{t}$) and salinity-driven component (halosteric change, hereafter

$Z_\mathrm{s}$), using the numerical formulation of (Tabata et al., 1986) which is given in pressure coordinates by

$$Z_\alpha = \frac{1}{g} \int\limits_{p_\mathrm{a}}^{p_\mathrm{b}} (\Delta\alpha)\,dp \tag{2a}$$

$$Z_\mathrm{t} = \frac{1}{g} \int\limits_{p_\mathrm{a}}^{p_\mathrm{b}} \left(\frac{\partial\alpha}{\partial T}\right) \Delta T\,dp \tag{2b}$$

$$Z_\mathrm{s} = \frac{1}{g} \int\limits_{p_\mathrm{a}}^{p_\mathrm{b}} \left(\frac{\partial\alpha}{\partial S}\right) \Delta S\,dp \tag{2c}$$

where $\alpha$ (m$^3$kg$^{-1}$) is the specific volume; $T$ (°C) is temperature, $S$ (gkg$^{-1}$) is salinity; $\Delta T = T - \bar{T}$ and $\Delta S = S - \bar{S}$ represent the mean monthly departure of $T$ and $S$ from their respective climatological annual means ($\bar{T}$ and $\bar{S}$); $\Delta\alpha$ is the departure of

specific volume corresponding to small values of $\Delta T$ and $\Delta S$; and the integration is carried out between pressure levels from the ocean surface to the bathymetry. Figure 5 shows the comparison of $Z_\alpha$ obtained from method one and two and the agreement between them is good, which gives us confidence in our calculations. One source of error in Eq. (2) is that, because we neglect higher order derivatives in the estimation of the thermal expansion coefficient ($\partial\alpha/\partial T$) and haline contraction coefficient ($\partial\alpha/\partial S$), Eq. (2) may not capture high frequency steric fluctuations. However, because our focus here is on long-term low

frequency $SLA$ fluctuations, this error is unlikely to affect our results.

The satellite Gravity Recovery and Climate Experiment (GRACE) data set provides monthly estimates of $G$ at 1° spatial resolution (Save et al., 2016; Save, 2020), but the record is short (March 2002 to October 2017) and it has many time gaps. To overcome this deficiency we used ORAS5 $G$ (see calculation details in the next section). Figure 2[b] shows that ORAS5 $G$ captures the GRACE $G$ trend pattern, which gives us confidence in our calculations.

It is also possible to estimate $G$ from its constituent components. Introducing $P$ (cm) for precipitation averaged over the domain, $E$ (cm) for evaporation averaged over the domain, $R$ (cm) for land runoff into the domain, and $F_{\mathrm{net}}$ (cm) for seawater net flux through the domain boundaries, then $G$ is given by

$$G = P - E + R + F_{\mathrm{net}} \tag{3}$$



In reality, however, it is difficult to calculate $F_{\text{net}}$ in Eq. (3) from ocean reanalysis data sets by calculating the fluxes through the domain boundaries because ORAS5 and most ocean reanalysis systems do not conserve mass. Therefore, because local

salinity and temperature (e.g., from Argo floats) are assimilated into ORAS5, which enhances its reliability, we used the calculated ORAS5 $G$ for the analysis here. By substituting this calculated $G$ and the estimated $P$, $E$ and $R$ into Eq. (3), we derived $F_{\text{net}}$ as a residual.

## 2.2   Data sets and processing

The altimeter $SLA$ measurements that we used is the climate-oriented gridded, monthly, 0.25° horizontal resolution, Coper-

nicus Climate Change Service satellite observations dataset version vDT2021, which is available from 1993 to present (C3S Climate Data Store, 2018). This satellite record is designed for monitoring the long-term evolution of sea level and other ocean and climate indicators, thus it is suitable for this study.

We obtained ocean reanalysis sea level anomaly ($SLA_{ORAS5}$), seawater salinity ($S$) and seawater temperature ($T$) from the monthly ECMWF ORAS5 ocean reanalysis data set (Zuo et al., 2019), which is available from 1979 to 2018. ORAS5 has

0.25° horizontal resolution and 75 vertical levels, and we downloaded it from the Integrated Climate Data Center, Hamburg University.

Notice the discrepancy between $SLA$ and $SLA_{ORAS5}$ (Fig. 2[a]): this is likely because 1) observations near the coast that are assimilated into ORAS5 have larger errors in general, 2) ORAS5 does not assimilate altimeter $SLA$ near the coast, and 3) vertical land displacement is not well represented in ORAS5 (Zuo et al., 2019). Compared to $SLA$, the period two minus

period one $SLA_{ORAS5}$ difference for the domain and subdomains are about 5% larger: this discrepancy is accounted for by the barometric and Boussinesq corrections (see Table A1). ORAS5 assimilates in situ and satellite measurements ($S$, $T$ and $SLA$), it uses information on the global mean sea level trend to close the fresh-water budget, and it includes a bias correction scheme for pressure in the tropical region and for salinity and temperature in the extra-tropical region (Zuo et al., 2019; Balmaseda et al., 2013). These ORAS5 characteristics are likely why $SLA_{\text{ORAS5}}$ long-term trend in this domain is in good agreement with

altimeter $SLA$ measurements (Fig. 2[a]). Overall, Carton et al. (2019) showed that ORAS5 is suitable for long-term variability studies and Ibrahim and Sun (2022) have verified ORAS5 ensemble control member (opa0), which we use here, in this domain. This gives us confidence that ORAS5 data set is suitable for this study.

To calculate ORAS5 $G$, first, we used ORAS5 $S$ and $T$, together with the Gibbs SeaWater (GSW) Oceanographic Toolbox of TEOS-10 version 3.6.19 in Python, to obtain in situ density (kg m$^3$); second, using this density, together with ORAS5 layer

thickness (m) and domain area (m$^2$), we calculated mass (kg); third, using seawater density of 1025 kg m$^{-3}$ and the domain area, we obtained mass in equivalent water thickness units (cm), the same units as GRACE data (Save et al., 2016). Note that ORAS5 does not assimilate GRACE. Accordingly, GRACE is an independent source to validate our calculations of ORAS5 $G$ and the agreement is good, as shown in Fig. 2[b].

We estimated $P$ minus $E$ ($P \text{minus} E$) and $\zeta_{\text{a}}$ using the monthly ECMWF ERA5 atmospheric reanalysis surface data set

(Hersbach et al., 2020), which is available from 1979 to present. ERA5 has 30 km spatial resolution and we downloaded it from the Copernicus Climate Change Service Climate Data Store (Hersbach et al., 2023). To estimate $R$ we calculated $P \text{minus} E$



over the catchment of the rivers that discharge into the domain (World Bank, 2019) (Fig. 1[b]). Compared to measured land runoff, this $R$ is likely more accurate since it also includes coastal groundwater discharge into the domain. The large land area extending inland around 10°N (Fig. 1[b]) is the catchment of the Niger River, which has its headwaters in the Fouta Djallon
Highland and flows inland. There are three other land areas near the coast, around 11°N, 13°N, and 15°N, that are unresolved in the rivers data set that we used (World Bank, 2019), probably because these areas have no guaged rivers. In general, river contribution to $SLA$ shift is comparatively small (Fig. 4[d]).

### 2.3 Analysis subdomains

As a first step to comparing the causes of the shift in $SLA$ trend pattern between period one and two, we carried out an
empirical orthogonal function (EOF) analysis of $SLA$ in the domain to identify coherent horizontal structures (Fig. 3). This is useful to ensure correct spatial averaging of atmosphere and ocean variables for time series analysis. EOF analysis mode 1 with the annual cycle removed explains more than 87% of the temporal-horizontal variance and it reveals two subregions with differing spatial patterns of $SLA$ variability (Fig. 3[b1]), which are hereafter denoted 'subdomain A' and 'subdomain B' as shown in Figure 3[b1,c1]. EOF mode 1 with the annual cycle removed accounts for a large proportion of the total variance
(87.3%), indicating that the spatial structure of $SLA$ variation revealed in mode 1 is stable (Fig. 3[b1]). EOF mode 2 with the annual cycle removed accounts for only 4% of the temporal-horizontal variance (Fig. 3[c1]). The horizontal pattern of mode 1 indicates a large $SLA$ difference (greater than 24 cm) between subdomains A and B Figure 3[b1].

The boundary between subdomain A and B is around the 14.75°N line of latitude, at the western tip of the African continent near Dakar, Senegal (Fig. 1). Subdomain A and B correspond to regions of comparatively shallow and deep bathymetry (Fig.
1[a]) as well as to regions of comparatively low and high mean sea surface height (Fig. 1[b]), respectively. Based on this EOF analysis, we analyze the causes of the shift in $SLA$ pattern in each subdomain between period one and two.

## 3 Multidecadal shifts in the causes of sea level fluctuations

In order to specify the drivers of the sea level rise hiatus, in section 3.1 we first describe the relative contributions of the three causes of sea level fluctuations (Eq. 1) to the multidecadal $SLA$ shift, i.e., steric, mass, and barometric shifts. In section 3.2 we
characterize the dominant driver of the hiatus.

### 3.1 Relative contributions of steric, mass, and barometric shifts to the sea level shift

The two period difference (i.e., period two minus period one) of the area-average multi-year mean $SLA$ and the factors that causes it to shift are given in Table A1, while we summarize the percentage contributions here. Using the residual calculation approach based on Eq. (1), the contribution of $Z_\alpha$ to the multi-year mean $SLA$ shift is $\approx 74.3\%$ in the whole domain, 69.8%
in subdomain A, and 76.6% in subdomain B; the contribution of $G$ to the $SLA$ shift is $\approx 21.7\%$ in the whole domain, 24.6% in subdomain A, and 20.1% in subdomain B; and the contribution of $\zeta_a$ to the $SLA$ shift is 3.56% in the whole domain, 4.88% in subdomain A, and 2.84% in subdomain B (Eq. (2), Eq. (3), Table A1).





**Figure 3.** Empirical Orthogonal Function (EOF) analysis of the domain monthly $SLA$. (**a1**) Mode 1, annual mean not removed. (**b1**) Mode 1, annual mean removed. (**c1**) Mode 2, annual mean removed. (**a2–c2**) a1, b1 and c1 time series, respectively.





**Figure 4.** Monthly time series (annual cycle removed) of domain and subdomain characteristics, and multi-year mean shift (i.e., period two minus period one) in parenthesis. (**a**) Area-averaged $SLA_{\mathrm{ORAS5}}$ in the domain (blue dash line), subdomain A (orange dash-dotted line), and subdomain B (purple dotted line). (**b**) ORAS5 $G$, curves are as in a. (**c**) ERA5 precipitation minus evaporation ($P\,\mathrm{minus}\,E$) over the ocean, curves are as in a. (**d**) ERA5-derived land runoff, curves are as in a. (**e**) ORAS5 potential temperature at 50 m depth in the Guinea Dome region: 10°N–13.5°N, 21°W–30°W (see discussion in section 5). A 13-month low-pass (moving average) filter is applied to all the time series.




**Figure 5.** Comparison of total steric ($Z_\alpha$) shift during 1985–2018 obtained with two different methods: in method one $Z_\alpha$ is obtained (red curve) as a residual using Eq. (1), i.e., $SLA$ minus $G$, and in method two $Z_\alpha$ is directly calculated (blue curve) using Eq. (2).

The temporal pattern of the residual $Z_\alpha$ (Eq. 1) and the calculated $Z_\alpha$ (Eq. 2) are almost identical (Fig. 5). The small discrepancy between them is likely because, unlike the calculated $Z_\alpha$, the shift in $\zeta_a$ is included in the residual $Z_\alpha$; and, as stated before, we neglect second and higher order Taylor expansion terms for the steric expansion/contraction coefficients in Eq. (2) (Tabata et al., 1986).

In the domain, subdomain A and subdomain B, $PminusE$ contributed about 18.2%, 2.97%, and 26.1% (Ibrahim and Sun, 2023) of the multi-year mean $SLA$ shift, respectively; $R$ contributed about $-0.241\%$, $-1.50\%$ and 0.441%, respectively; and $F_{\text{net}}$ contributed $\approx 3.82\%$, 23.2% and $-6.45\%$, respectively, (Fig. 4 [c,d], Eq. (3), Table A1).





195 In summary (Table A1), the three dominant drivers of $SLA$ shift in the domain, in decreasing order of effect, are $Z_\alpha$, $PminusE$ and $\zeta_a$; for subdomain A they are $Z_\alpha$, $F_{net}$ and $\zeta_a$; and for subdomain B they are $Z_\alpha$, $PminusE$ and $F_{net}$. As expected, steric shift is the dominant driver of the observed sea-level variation pattern in the domain as well as in the two subdomains. The second dominant driver in subdomain A ($F_{net}$, 23.2%) is different from the second dominant driver in subdomain B ($PminusE$, 26.1%). Since $F_{net}$ expresses mass exchange between ocean regions and $PminusE$ expresses mass

200 exchange between the ocean and the atmosphere, one interpretation of these results is that, with regards to mass effects on long-term $SLA$ variability (i.e., $G$ in Eq. (1)) in this domain, oceanic processes predominate in subdomain A, while atmosphere-ocean processes predominate in subdomain B. This interpretation is consistent with the differing pattern of $SLA$ variability in subdomain A and subdomain B that is revealed in the EOF analysis (Fig. 3[b1]). Note that, because $F_{net}$ and $PminusE$ modulate the domain temperature and salinity (and therefore the domain density), these two factors also contribute to the steric

205 shift ($Z_\alpha$). To characterize the mechanism of $Z_\alpha$ and show the operating processes, in the following section we compare the relative roles of temperature and salinity on $Z_\alpha$.

### 3.2 Relative effects of temperature and salinity on steric shift

In order to assess the comparative roles of temperature and salinity on the multi-year shift in subdomains A and B $Z_\alpha$, and to specify the ocean current changes associated with the multi-year shift in subdomains A and B $F_{net}$, we examined subdomains A

210 and B vertical structure. Figure 6[a1,b1] and Figure 6[a2,b2] show the T-S profile and T-S diagram, respectively, in subdomains A and B, averaged over periods one and two. There is a decrease in subdomain A salinity in the depth range of $\approx$ 0–1265 m during period two (Fig. 6[a1]), and the associated change in density structure is also evident (Fig. 6[a2]). Near-surface salinity decreased in subdomain B during period two (Fig. 6[b2]), which confirms the multi-year mean increase in subdomain B $PminusE$ (Fig. 4[c]).

215 The change in subdomain B density structure is mainly between surface and $\approx$ 300 m. In subdomains A, period two shift in density vertical structure is dominated by salinity shifts (Fig. 6[a2]), suggesting that, compared to temperature, salinity adjustment plays a more important role in the multi-year upward shift in subdomain A $Z_\alpha$. To verify this and specify the temperature and salinity shifts temporal pattern at differing depths, we separated subdomains A and B into four vertical layers based on their salinity vertical profile (Fig. 7[a1,b1]). Layer 1 is at depth 0–50 m, layer 2 is at depth 50–735 m, layer 3 at depth

220 735–1725 m, and layer 4 is at depth 1725–5000 m.

 Figure 7 shows the layer-averaged monthly time series (annual cycle removed) of temperature and salinity in subdomain A and B. Compared to period one, period two subdomain A layers 1–3 temperature decreased by 0.164°C, 0.0133°C and 0.110°C, respectively, while layer 4 temperature increased by 0.0386°C (Fig. 7[a1]): consequently, only layer 4 thermosteric expansion contributed to the multi-year upward shift in subdomain A $Z_\alpha$. Subdomain A layers 1–4 salinity decreased by 0.0653 g/kg,

225 0.0211 g/kg, 0.0408 g/kg and 0.00122 g/kg, respectively, between periods one and two (Fig. 7[a2]): consequently, halosteric expansion in the entire water column contributed to the multi-year upward shift in subdomain A $Z_\alpha$. Between periods one and two subdomain B layers 1–4 temperature increased by 0.0488°C, 0.219°C, 0.0619°C and 0.0181°C, respectively (Fig. 7[b1]), implying that thermosteric expansion in all four layers contributed to the multi-year upward shift in subdomain B $Z_\alpha$.

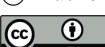



**Figure 6.** Multi-year mean vertical structure in the domain. (**a1, b1**) Period one (dotted line) and period two (solid line) T-S profile for subdomains A and B, respectively; salinity is shown in blue and temperature is shown in red. (**a2, b2**) Period one (red dotted line) and period two (blue dotted line) T-S diagram for subdomains A and B, respectively. Notice the large period two shift in subdomain A salinity (a1) and density (a2).





Compared to period one, period two subdomain B layer 2 salinity increased by 0.0284 g/kg, while layers 1, 3 and 4 salinity

decreased by 0.0962 g/kg, 0.00501 g/kg and 0.00209 g/kg, respectively (Fig. 7[b2]), implying that layers 1, 3 and 4 halosteric expansion contributed to the multi-year upward shift in subdomain B $Z_\alpha$.

To ascertain the comparative role of temperature and salinity in subdomain A and B steric shifts, we calculated the thermosteric ($Z_t$) and halosteric ($Z_s$) shifts in each subdomain (Eq. 2b and 2c, Fig. 8). In the domain as a whole, $Z_t$ is slightly larger and $Z_s$, 1.51 cm versus 1.09 cm (Fig. 8), respectively. However, there are large differences between the two subdomains.

In subdomain A, $Z_t$ is $\approx -1.02$ cm and $Z_s$ is $\approx 3.28$ cm; while in subdomain B $Z_t$ is $\approx 3.07$ cm and $Z_s$ is $\approx -0.252$ cm (Fig. 8). This means that in the domain, $Z_t$ and $Z_s$ contributed about 58% and 42%, respectively, of the total steric shift ($Z_\alpha$); in subdomain A, $Z_t$ and $Z_s$ contributed about $-45\%$ and 145%, respectively, of $Z_\alpha$; and in subdomain B $Z_t$ and $Z_s$ contributed about 109% and $-9\%$, respectively, of $Z_\alpha$. Therefore, halosteric expansion is the dominant driver of the multi-decadal upward steric shift in subdomain A, while thermosteric expansion is the dominant driver of the upward steric shift in subdomain B.

These results highlight the role of salt as an important driver of long-term regional sea level shift, a role that is somewhat less recognized (compared to the role of heat) in discussions of sea level shifts under global climate change (Durack et al., 2014). Indeed, Ponte et al. (2021); Llovel and Lee (2015); Antonov et al. (2002) have suggested that shifts in salinity affect $SLA$ shift in large areas of the world ocean at short and long timescales.

It is beyond the scope of this work to analyze the domain heat and salt balance to find the physical processes that caused

the thermosteric and halosteric shifts in subdomain A and B. However, owing to three peculiarities of subdomain A that suggest the source of the freshening that caused the large halosteric shift there (145%/3.28 cm, Table A2), we give a probable explanation for subdomain A freshening. First, $P$ minus $E$ contributed only 2.97%/0.0962 cm of subdomain A $SLA$ shift, indicating that precipitation is not the key driver of subdomain A freshening. Second, river runoff inside subdomain A is small ($-1.50\%/-0.0486$ cm) and there is no river runoff north of it, indicating that freshwater from land is not the source of

subdomain A freshening. Third, the Mediterranean Sea outflow at the Strait of Gibraltar north of subdomain A has a higher salinity than subdomain A (Fig. 9[a1,b1,c1]) (Baringer and Price, 1997; Naranjo et al., 2017; Aldama-Campino and Döös, 2020), indicating that this outflow is not the cause of subdomain A freshening. Accordingly, we hypothesized that subdomain A freshening associated with the multi-year strengthening of $F_{net}$ has its origin in salinity shifts occurring elsewhere in the North Atlantic. To verify this hypothesis, we find the ocean currents that freshened subdomain A and describe them in the next

section.

## 4 Freshening of the Canary Current by subpolar North Atlantic water mass

The most prominent ocean current traversing subdomain A is the southward-flowing Canary Current, a relatively shallow (down to $\approx 200$–300 m) eastern boundary current of the North Atlantic Subtropical Gyre conveying North Atlantic Central Waters (Zenk et al., 1991; Mason et al., 2011; Pelegri and Peña-Izquierdo, 2015). The waters feeding the Canary Current

comes from the subpolar North Atlantic area, partly through the so-called 'Azores current,' a relatively deep (down to $\approx 2000$ m) narrow zonal jet that originates in the western North Atlantic margin where the Gulf Stream bifurcates. The Azores current





**Figure 7.** Monthly time series (annual cycle removed) of area-and-layer average ocean properties. (**a1**) Subdomain A potential temperature (temp.). (**a2**) Subdomain A salinity. (**a3**) Subdomain A salinity versus subpolar North Atlantic (52°N–53°N, 46°W–47°W) salinity, both averaged in layers 2 and 3; a 13-month low-pass (moving average) filter is applied to the subpolar time series to remove high frequency fluctuations that are attenuated during transit to the domain. (**b1**) Subdomain B potential temp. (**b2**) Subdomain B salinity. The numbers in parenthesis are period two minus period one values of temperature and salinity in each layer.





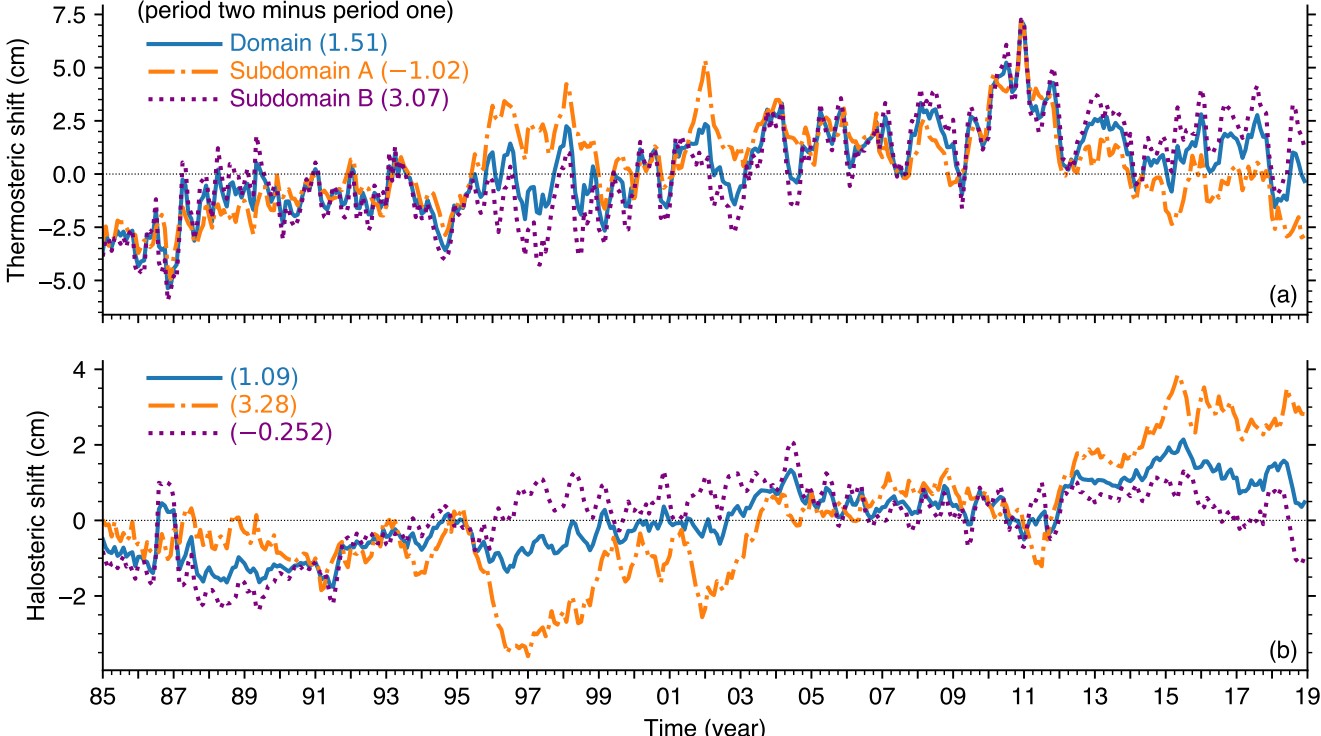

**Figure 8.** Monthly steric characteristics. (**a**) Area average thermosteric shifts in the domain (blue), subdomain A (orange) and subdomain B (purple). (**b**) Same as a but for halosteric shifts: notice the comparatively large multi-year halosteric shift in subdomain A. The time series are obtained from direct calculations using Eq. (2). The sum of **a** and **b** is the total steric shift, $Z_\alpha$, which are given in Table A2.

propagates southward and then eastward between 33°N and 35°N towards the African coast and the Gibraltar Strait (Klein and Siedler, 1989; Stramma and Müller, 1989; Jia, 2000; Comas-Rodríguez et al., 2011; Mason et al., 2011; Frazão et al., 2022). Figure 9[a1,b1,c1] shows the domain horizontal currents and salinity averaged in layers 1, 2 and 3, respectively, and Figure 265 7[a2,b2] show the time series of mean monthly salinity anomaly averaged in these layers for subdomains A and B. The linkage between the Azores current, which turns southward around 32°N, and the Canary Current, which traverses subdomain A, is particularly evident in layers 1 and 2 (Fig. 9[a1,b1]). Compared to period one, in period two the Canary Current strengthened by ≈ 0.01 m/s in northern subdomain A where salinity decreased by up to 0.4 g/kg (Fig. 9[a2]).

To find and describe the physical linkage between subdomain A freshening and salinity shifts occurring elsewhere in the 270 North Atlantic, we calculated and analyzed the maximum Spearman cross-correlation (during 1985–2018) at different time lags (1 to 60 months with subdomain A salinity lagging) and the linear regression coefficient (during period two) between salinity averaged in subdomain A layers 2 and 3 (dependent variable) and salinity at each grid-point in the North Atlantic averaged in layers 2 and 3 (explanatory variable) (Fig. 10). We chose these layers because, first, they have comparatively large





**Figure 9.** Salinity and ocean current structures and their multi-year shifts. (**a1,b1,c1**) Multi-year (period one) mean salinity and vector currents in layers 1–3, respectively. (**a2,b2,c2**) The same as a1–c1 but for period two minus period one. Notice the Guinea Dome indicated by the cyclonic circulation region between 8°N and 18°N in a1.

salinity shifts in subdomain A during period two (Fig. 7[a2]); second, they account for 71.6% of the total steric expansion in
subdomain A (not shown); and third, short-term air-sea fluxes that modify salinity are mostly attenuated in layers 2 and 3.

The spatial structure of the maximum cross-correlation (Fig. 10[a]) and regression coefficient (Fig. 10[b]) reveal two pathways whereby western subpolar North Atlantic waters propagate to the source area of the Canary Current. Path 1 (green colored line) is consistent with the Azores current: western subpolar waters propagate southward and then eastward between 33°N and



35°N towards the African coast. Conversely, in path 2 (cyan colored line) western subpolar waters propagate eastward to the
north European coast and then southward along the coast to subdomain A (Fig. 10). Moreover, time and depth average currents
suggest that path 1 waters partly come from path 2 in a westbound flow around the 45° N line of latitude (thin green colored
line) (Fig. 11[a]). We conclude therefore, based on path 1 and path 2, that there is a physical linkage between subdomain A
salinity and salinity in the western subpolar North Atlantic.

Since it takes time for the currents carrying subpolar North Atlantic water mass to travel along path 1 and 2 to the west
African coast, a necessary precondition for subpolar waters to drive subdomain A freshening is to observe subpolar North
Atlantic freshening prior to subdomain A freshening. Accordingly, based on the maximum correlation of salinity spatial pattern
(Fig. 10[a]), we compared the mean, monthly salinity anomaly time series for subdomain A and for a representative subpolar
North Atlantic region (46°W–47°W, 52°N–53°N, purple colored box in Fig. 10[a]), both calculated in the same ocean layer (2
and 3). Neglecting short-term fluctuations and focusing on the long-term trend, the negative salinity anomaly in subdomain A
began around 2004, while in the subpolar region the negative salinity anomaly began around 1999 (Fig. 7[a3]), demonstrating
that the subpolar freshening preceded subdomain A freshening.

To show that fresher subpolar water mass is the driver of subdomain A freshening, thus confirming our hypothesis, we
calculated and analyzed the potential vorticity, a conservative flow tracer that characterize ocean water mass. Introducing $f$
(1/s) for the planetary vorticity, $\zeta$ (1/s) for the relative vorticity, and $H$ (m) for the difference between the depth of two density
surfaces (1025 kg/m$^3$ and 1037 kg/m$^3$), the potential vorticity, $PV$, is given by (Apel, 1987)

$$PV = \frac{f+\zeta}{H} \tag{4}$$

We chose these two density surfaces based on the density boundaries between layer 1 and 2, and between layer 3 and 4.
Conservation of potential vorticity implies that, away from boundaries and the sea-surface, the ratio of the quantities on the
right-hand side of Eq. (4) is conserved following the motion of a water mass parcel. Thus, variations in $PV$ show variations in
the associated water mass. To compare period one $PV$ ($PV_{\text{period 1}}$) and period two $PV$ ($PV_{\text{period 2}}$) while preserving the $PV$
pathway, we used a $PV$ ratio metric given by

$$PV \text{ ratio} = \left( \frac{PV_{\text{period 1}}}{PV_{\text{period 2}}} - 1 \right) \times 100 \tag{5}$$

Eq. (5) means that when $PV_{\text{period 1}}$ is greater than $PV_{\text{period 2}}$, $PV$ ratio is positive; and when $PV_{\text{period 1}}$ is less than
$PV_{\text{period 2}}$, $PV$ ratio is negative. Figure 11[a,b] shows layers 2 and 3 averaged currents and the $PV$ ratio calculated be-
tween these depths, respectively. The subpolar North Atlantic water mass propagation along **path 1** and **path 2** is visually clear
in the $PV$ ratio spatial pattern (Fig. 11[b], Fig. 10[a]). The positive $PV$ ratio along these two paths imply that, compared to
period one, $PV$ is lower in period two. We conclude therefore that subdomain A freshening during period two is because of
the Canary Current freshening by currents carrying fresher subpolar North Atlantic water mass via path 1 and path 2.

The maximum lagged correlation between the time series of subpolar North Atlantic salinity and subdomain A salinity, $\approx$
0.687, occurred when subdomain A salinity is lagging by about 65 months, which is consistent with Figure 7[a3]. Hence, the
transit period to propagate the fresher subpolar waters to subdomain A is about 5.5 years. We do not know the cause of the





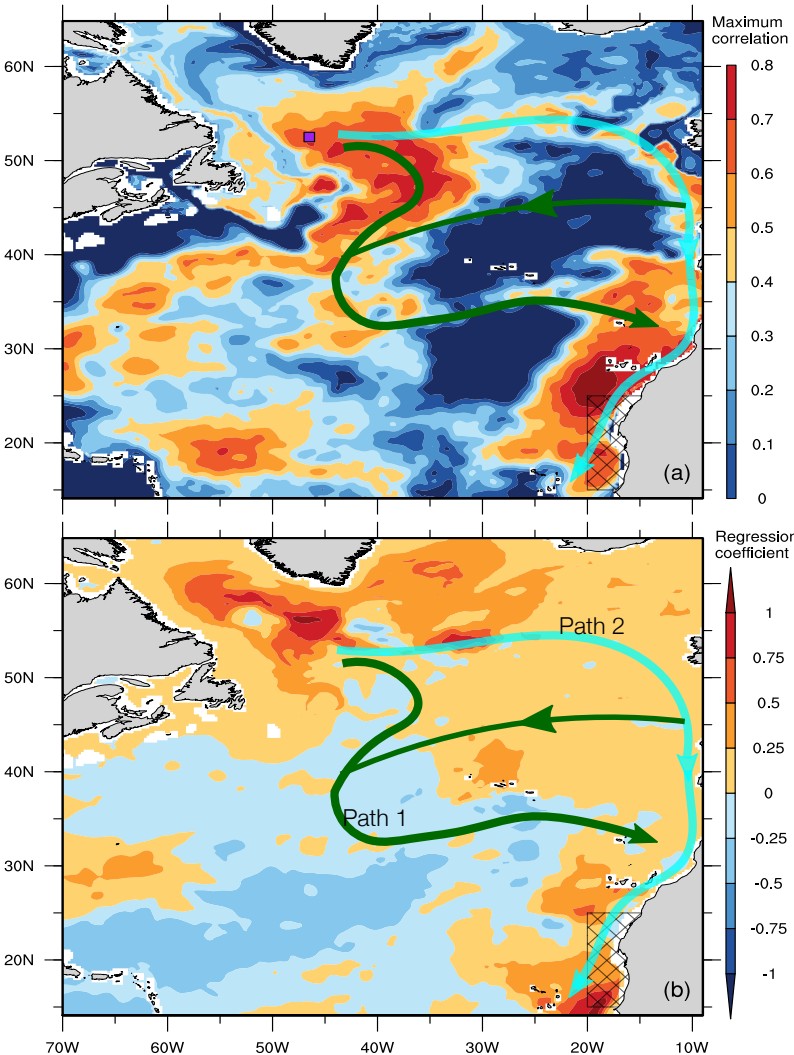

**Figure 10.** Physical linkage between subdomain A salinity and salinity elsewhere in the North Atlantic. (**a**) Maximum spearman cross-correlation during 1985–2018 at different time lags (1 to 60 months) between salinity averaged in subdomain A layers 2 and 3 and salinity averaged in layers 2 and 3 at each point in the North Atlantic, with subdomain A salinity lagging; the hatched region is subdomain A and the purple colored box (46° W–47° W, 52° N–53° N) is the high correlation region chosen to represent the subpolar North Atlantic. (**b**) Period two linear regression coefficient between subdomain A salinity (dependent variable) and salinity averaged in layers 2 and 3 at each point in the North Atlantic (explanatory variable). Notice the large correlation and regression coefficient between subdomain A salinity and salinity in the subpolar North Atlantic. The curves overlayed on a and b show the probable pathways whereby subpolar waters propagate to subdomain A. In **path 1** (green curve) subpolar waters flow southward to around 30°N line of latitude and then flow eastward towards the northwest African coast where the Canary Current (that traverse subdomain A) originates; path 1 is consistent with the course of the so-called Azores current. In **path2** (cyan curve) subpolar waters flow eastward along about 52°N line of latitude to the northern European coast and then flow southward along the coast to subdomain A; a westward flow connects path 2 and path 1 around the 45°N line of latitude (thin green curve), see potential vorticity and layer-averaged velocity in Fig. 11.



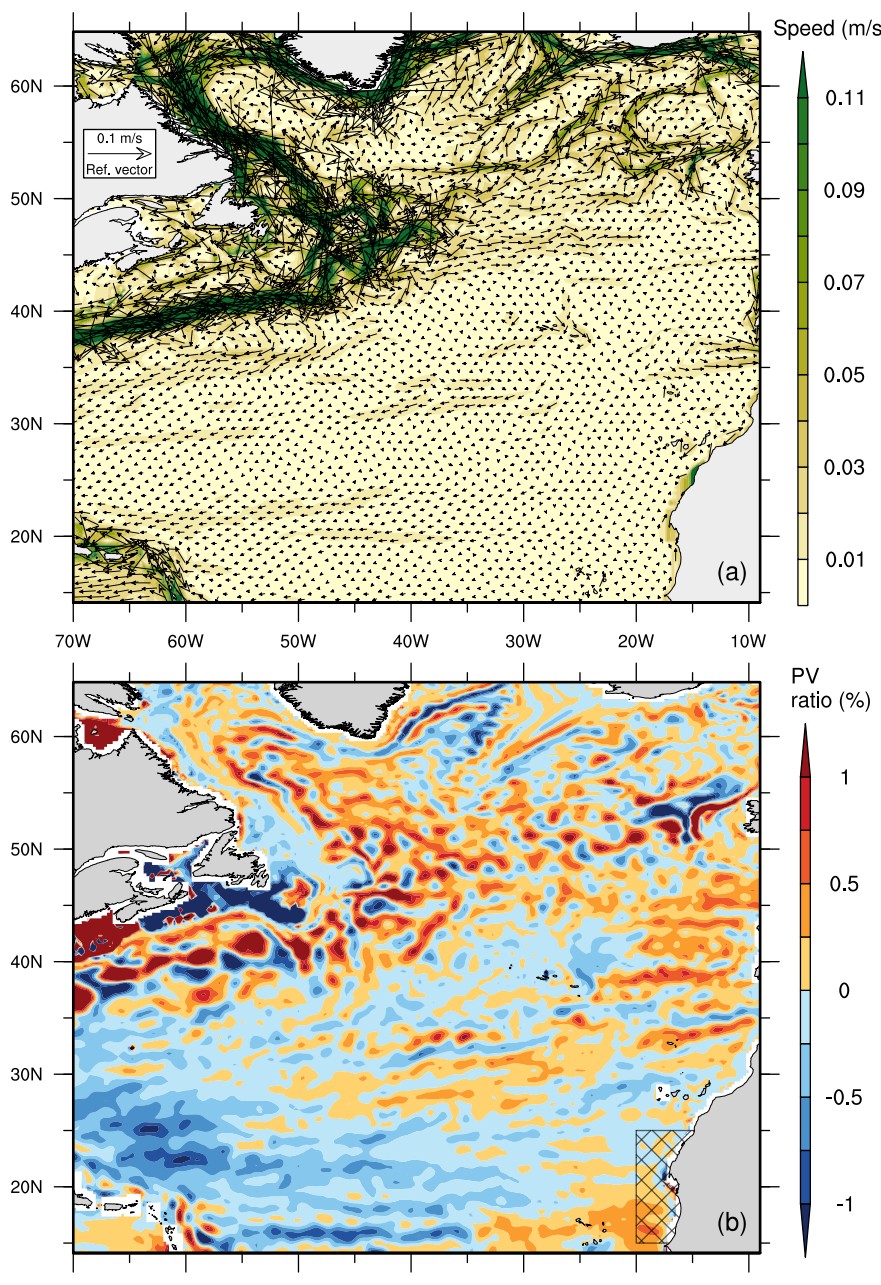

**Figure 11.** Physical relation between subdomain A salinity shift and salinity shift elsewhere in the North Atlantic. (**a**) Time (1985–2018) and layer (2 and 3) average currents. (**b**) $PV$ ratio (Eq. 5) calculated between the depths of density layer 1025 kg/m$^3$ and and density layer 1037 kg/m$^3$, i.e., layers 2 and 3, see Eq. (4). Notice the positive $PV$ ratio ($PV$ decrease in period two) along **path 1** and **path 2** whereby subpolar North Atlantic water mass propagates to northern subdomain A.



freshening in the subpolar North Atlantic. Holliday et al. (2020) reported that, owing to unusual winter wind patterns that rerouted Arctic-origin freshwater, in 2011 the largest freshening event in 120 years occurred in the subpolar North Atlantic area. However, the time series of salinity anomaly in the subpolar North Atlantic shows that the freshening there began as far back as late 1995 (Fig. 7[a3]). There is need for further research to characterize the salt dynamics in this subpolar region.

     In summary, the results here show that a key aspect of the domain sea level rise hiatus is the freshening of subdomain A by
fresher subpolar North Atlantic waters conveyed to the West African coast along two different paths, an open ocean path (path 1) and a coastal path (path 2). This work, accordingly, highlights a multidecadal linkage between salinity anomalies occurring in the western subpolar North Atlantic and sea level anomalies occurring in the eastern tropical North Atlantic. Moreover, the currents flowing along path 1 and path 2 also bring subpolar waters to the source area of the Mediterranean inflow near the Strait of Gibraltar (Fig. 9[a1,b1,c1], Fig. 10), suggesting the possibility of this same multidecadal linkage between the
western subpolar North Atlantic and the Mediterranean Sea. Indeed, analysis of numerical experiments by (Jia, 2000) and (Özgökmen et al., 2001) suggest that the emergence of the Azores current is associated with the water sink generated in the eastern boundary by entrainment of the Mediterranean outflow as it descends the continental slope and by the Mediterranean inflow. Experimental research is needed to classify the modes of variability of currents along path 1 and path 2 and its linkage to ocean dynamics in the eastern boundary of the North Atlantic.

## 5    Guinea Dome thermal structure shift and water accumulation

Mass shift, $G$ (Eq. (3), Fig. 4[b], Table A1), contributed more than 20% of the multi-year sea level shift in subdomains A and B, implying water accumulation inside the domain. Accordingly, we hypothesized a shift in ocean currents structure that enhanced domain-wide water accumulation.

     In the area west of subdomain B, layer 1 averaged horizontal currents shows a cyclonic (anticlockwise) circulation region
centered around 12°N and 22°W (Fig. 9[a1], Fig. 12). This is the so-called 'Guinea Dome,' a permanent, quasi-stationary feature situated between the westward-flowing North Equatorial Current and the eastward-flowing North Equatorial Countercurrent. This thermal dome is characterized by upward flux of cooler water from beneath that causes doming (upward displacement) of isotherms (Mazeika, 1967; Siedler et al., 1992; Yamagata and Iizuka, 1995; Doi et al., 2009, 2010).

     To verify the hypothesis that a shift in ocean currents structure enhanced domain-wide water accumulation (domain-wide
increase in $G$ contributed $\approx 22\%$ of the $SLA$ shift, Table A1), our approach is to first demonstrate a linkage between horizontal velocity in the domain and vertical velocity in the Guinea Dome region. From considerations of the continuity of water mass, in order for the displaced isotherms in the Guinea Dome to remain stationary, divergent (convergent) flow in the layer above the dome must be coupled to convergent (divergent) flow in a subsurface layer. Wyrtki (1964) offers an instructive illustration of the water balance of a thermal dome: consider a circular region of radius $r$ (m) below the surface surrounding the dome core,
the water flowing upward (downward) through this surface, with vertical average velocity $w$ (m/s), must be removed (replaced) by horizontal water flow, with average velocity $u$ (m/s), in a layer of thickness $h$ (m) overlying the surface, thus

$$u \cdot 2\pi r h = w \cdot \pi r^2 \tag{6}$$



**Figure 12.** Thermal structure at 50 m depth, where the Guinea Dome (GD) is better developed (Mazeika, 1967; Doi et al., 2009) and its multi-year mean shift. (**a1,a2,a3**) season one (March–October) mean potential temperature in period one, period two, and period two minus period one. (**b1,b2,b3**) the same as a1–a3 but for season two (November–April). Notice the warming between 10°N and 15°N.

Eq. (6) shows that a change in vertical velocity in the dome core is associated with a change in horizontal velocity in a layer above the dome. Because the displaced isotherms do not reach the sea-surface (Doi et al., 2009), which demonstrates that the dome circulation must be in thermal balance, the maximal $w$ is thus limited by the heat energy available for warming the water from beneath as it ascends (Wyrtki, 1964). Accordingly, a shift in this available heat energy implies a shift in $w$, and consequently, a shift in $u$ in accordance with Eq. (6).

We now describe the mutual shift in the dome thermal structure and water accumulation in the domain. A useful index of the dome thermal balance is the 50 m temperature in the dome core (Mazeika, 1967). The multi-year mean shift in layer 1 currents reveal an anticyclonic (clockwise) circulation region north of the Guinea Dome center around 12°N–16°N and 22°W–28°W (Fig. 9[a2]). This circulation will cause a convergence zone and the associated downwelling of surface water will warm





the lower layers, as demonstrated by the shift in 50 m temperature monthly time series (Fig. 4[e]) and seasonal contour (Fig. 12[a3,b3]) during period two. The downwelling, moreover, is opposite to the permanent upward flux of water in the Guinea Dome, which implies a weakening of the dome circulation between periods one and two. Our explanation, therefore, for the domain-wide water accumulation is that the necessary adjustment of horizontal current velocities (to the shift in Guinea Dome

thermal balance) strengthened period two surface flow towards the convergence zone from subdomains A and B (Fig. 9[a2]), thereby enabling water accumulation in both of them.

The above explanation suggest an interrelated pattern of multi-year shift in several permanent ocean elements operating in this region. Indeed, Cromewell (1958) suggested that the Costa Rica Dome in the eastern tropical Pacific Ocean, which exists in a similar topographic setting as the Guinea Dome (i.e., located in the northern edge of the North Equatorial Countercurrent

and in close proximity to continental land-masses, in the eastern ocean boundary), is associated with cyclonic shear arising from the interaction of the countercurrent and a northward coastal current. There is a year-round northward coastal current off the northwest African coast (Ibrahim and Sun, 2022) and Figure 9[a2] shows a multidecadal shift in the horizontal currents in the northern edge of the countercurrent between 8°N and 9°N lines of latitude. It is thus probable that the domain-wide water accumulation, the shift in the dome vertical velocity and thermal structure, and the shift in horizontal currents are all associated

with the same phenomena occurring at multidecadal timescale.

One hypothesis that Wyrtki (1964) proposed for shifts in the Costa Rica Dome is that the cyclonic circulation separating the North Equatorial Current and Countercurrent (just as in Fig. 9[a1]) is unstable, probably because of fluctuations in the countercurrent strength and transport, which leads to recurring separation of eddies, and consequently, to shifts in the thermal dome structure. Further research is needed to obtain estimates of the timescale for which eddies separate from tropical thermal

domes since it may be associated with sea level fluctuations in the tropical ocean margins. In situ under water observations, for example, can be useful for elucidating the annual variation pattern of thermal dome spatial structure in the ocean margins.

## 6   Concluding remarks

We investigated the multidecadal sea level rise hiatus in the eastern tropical Atlantic margin off northwest Africa by comparing the drivers during 2010–2018, the hiatus period, and during 1996–2004, when the sea level there was rising (2[a]). These are

our major findings and interpretation.

1) In the domain as a whole, seawater expansion (steric shift) associated with shifts in density structure contributed about 74% of the multi-year upward sea level shift. Temperature-driven steric shift (thermosteric expansion) is dominant in the southern subdomain, while salinity-driven steric shift (halosteric expansion) is dominant in the northern subdomain, although thermosteric expansion here is also large (section 3). This halosteric expansion is associated with freshening of

the Canary Current that traverses the northern subdomain (section 4). The Canary Current source area was freshened by western subpolar North Atlantic waters that reached the northwest African coast via two pathways: an open-ocean path that is consistent with the Azores current, and a Western Europe coastal ocean path. We estimate the time to propagate this freshening to the northwest African coast to be about 5.5 years.





2) These results emphasize the role of salt as a key driver of long-term regional sea level shift, a role that is somewhat not well recognized in the literature. This is especially important in the context of the globally changing climate because changes in local hydrological cycles can drive large shifts in salinity, resulting in large shifts in regional sea level.

3) Another conclusion from these results is a multidecadal linkage between salinity anomalies in the western subpolar North Atlantic and coastal sea level anomalies in the eastern tropical North Atlantic. It is thus probable that, given satellite or in situ observations of salinity shifts in this subpolar region, coastal sea level shifts in the northwest African coast can be anticipated with a lead time of about 4 years. The subpolar North Atlantic waters flowing in the open-ocean and coastal paths also supply water to the Mediterranean inflow region near the Strait of Gibraltar (Fig. 9[a1], Fig. 10, Fig. 11 section 4). Accordingly, it may also be possible to anticipate multidecadal shifts in the Mediterranean Sea characteristics based on observed shifts in the subpolar North Atlantic. Because freshening in the subpolar North Atlantic region is associated with surface phenomena (precipitation, river runoff, ice melt), surface observations may be useful for estimating this lead time. It is interesting and useful to contrast this low-frequency remote forcing of sea level in the ocean margin, which is realized through shifts in the salinity of source water advected to the margin, with high-frequency local atmospheric forcing that is realized through shifts in surface pressure over the margin. Considerations of the timescale and magnitude of sea level shift associated with these two forcing types facilitate design of sustainable infrastructure in coastal regions.

4) Domain-wide water accumulation contributed about 22% of the multi-year upward sea level shift. In the southern sub-domain, the upward shift in precipitation contributed about 26% of the sea level shift there, whereas in the northern subdomain the upward shift in the net flux of seawater into this region contributed 23% of the upward sea level shift (Table A1, section 3). The dynamical adjustment of horizontal currents that enabled this water accumulation pattern appear to be related to multi-year shifts in the characteristics of several permanent ocean elements operating in this tropical Atlantic region including the Guinea Dome thermal structure and vertical velocity, and horizontal currents in the northern edge of the North Equatorial Countercurrent (section 5).

5) It is difficult to isolate the underlying mechanisms of this interrelated multidecadal shift in key elements of the ocean and atmosphere circulation operating this ocean margin (Fig. 1) by analyzing observational reanalysis data sets, as we have done here. This work thus highlight an important need for a high-resolution atmosphere and ocean coupled model to incorporate all current knowledge in this ocean margin, to provide a system for performing numerical experiments that will increase our understanding of the operating mechanism in this vitally important tropical region, and to facilitate better predictions. This will not only benefit neighboring countries, but it will improve our overall understanding of ocean-atmosphere interactions in ocean margins.





## Appendix A: Shift in sea level anomaly ($SLA$) and in the driving atmosphere, ocean and land variables

**Table A1.** Table values show the contribution of atmosphere and ocean variables to shift in sea level anomaly ($SLA$) in the North Atlantic margin off northwest Africa. The values in parenthesis in column 2, 3 and 4 indicate the percentage contribution of each variable to the $SLA$ shift (row 1) in the domain, subdomain A, and subdomain B, respectively. $SLA = SLA_{\mathrm{ORAS5}}$+ barometric correction ($\zeta_a$) + Boussinesq correction (see discussion in method section). The total steric (residual $Z_\alpha$) shifts reported in row two is obtained using the residual calculation approach (Eq.1).

| Variable [cm] | Domain | Subdomain A | Subdomain B |
|---|---|---|---|
| Sea level anomaly ($SLA$) | 3.52 (100%) | 3.24 (100%) | 3.70 (100%) |
| ORAS5 $SLA$ ($SLA_{\mathrm{ORAS5}}$) | 3.37 (95.9%) | 3.06 (94.5%) | 3.57 (96.6%) |
| Total steric (residual $Z_\alpha$) | 2.61 (74.3%) | 2.26 (69.8%) | 2.83 (76.6%) |
| Mass ($G$) | 0.764 (21.7%) | 0.797 (24.6%) | 0.741 (20.1%) |
| Transport ($F_{\mathrm{net}}$) | 0.134 (3.82%) | 0.749 (23.2%) | −0.238 (−6.45%) |
| Rainfall/Evaporation ($P minus E$) | 0.638 (18.2%) | 0.0962 (2.97%) | 0.963 (26.1%) |
| River runoff ($R$) | −0.00844 (−0.241%) | −0.0486 (−1.50%) | 0.0164 (0.441%) |
| Boussinesq correction ($\varepsilon$) | 0.0211 (0.571%) | 0.0211 (0.621%) | 0.0211 (0.541%) |
| Barometric ($\zeta_a$) | 0.125 (3.56%) | 0.158 (4.88%) | 0.105 (2.84%) |

**Table A2.** Table values show the salinity-driven (halosteric) and temperature-driven (thermosteric) contributions to total steric shift obtained from direct calculations using Eq. (2). The values in parenthesis in column 2, 3 and 4 indicate the percentage contributions in the domain, subdomain A, and subdomain B, respectively.

| Variable [cm] | Domain | Subdomain A | Subdomain B |
|---|---|---|---|
| Total steric ($Z_\alpha$) | 2.60 | 2.26 | 2.82 |
| Thermosteric ($Z_t$) | 1.51 (58.1%) | −1.02 (−45.1%) | 3.07 (108.9%) |
| Halosteric ($Z_s$) | 1.09 (41.9%) | 3.28 (145.1%) | −0.252 (−8.90%) |

*Data availability.* All the data sets that we used for this study are publicly available and can be found with the following website links: 1) the GEBCO bottom topography data set: https://www.gebco.net/data_and_products/gridded_bathymetry_data/; 2) the satellite altimetry sea-level data set: https://doi.org/10.24381/cds.4c328c78; 3) the GRACE mass change data set: http://www2.csr.utexas.edu/grace; 4) the ERA5 monthly data sets: https://doi.org/10.24381/cds.f17050d7; 5) the ECMWF ORAS5 data set: https://www.cen.uni-hamburg.de/en/icdc/data/ocean/easy-init-ocean/ecmwf-oras5.html.

415



*Author contributions.* HDI and YS jointly conceived the study, analyzed the data, and wrote and reviewed the manuscript draft.

420 *Competing interests.* The authors declare that they have no conflict of interest and have not received any specific funding for this work.

*Acknowledgements.* We thank the following agencies that produced the data sets that we used: Jet Propulsion Laboratory of the National Aeronautics and Space Administration, GFZ German Research Center for Geosciences, Jet Propulsion Laboratory, the University of Texas Center for Space Research, Astrium GmBH, Space Systems Loral, Onera and Eurockot GmBH, European Center for Medium-Range Weather Forecast, European Space Agency, and and Copernicus Marine Environment Service.





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
