# Peer review of "Multidecadal sea level rise hiatus in the tropical Atlantic margin off northwest Africa"

_EGUsphere, 2025_

## Author Comment (AC1)

**Response to Reviewer 1**

Thank you for reviewing our manuscript and for the inciteful comments and suggestions. We have addressed these comments, as shown below, and with line numbers for reference to the revised manuscript. Our responses are in blue font below each comment.

This manuscript aims at understanding the causes of the multidecadal sea level rise hiatus that occurred in 2010-2019 in the eastern north tropical Atlantic margin off northwest Africa. To do so, it mainly uses the ocean reanalysis ORAS5, performs an empirical orthogonal function analysis to define subdomains, and decomposes the sea level anomaly into its different components.

This manuscript is interesting, of good scientific quality, well structured, and well-written making it easy to follow the stream of thoughts. It addresses the scientific question of better understanding the drivers of sea level variability in the margin off northwest Africa and how it could be linked to remote processes.

Thank you.

I have two major comments and a few minor remarks which I think should be addressed before publication.

Major comment:

My first major comment concerns the scientific question being asked and which is not aligned with the analyses and results. It seems to me that the analyses presented do not answer the question aimed by the authors (specified lines 43-45) which is to find why the sea level stopped rising over period 2. Instead, by comparing the multi-year mean sea level and its components between the two periods, the authors answer a different scientific question which is: Why is the sea level higher during period 2 than during period 1? The concluding remarks section and the results summarized in the abstract indeed highlight the causes of the higher SLA during period 2 compared to period 1: mostly a halosteric expansion in subdomain A and a thermosteric expansion in subdomain B. The mass of the domain is also greater during period 2 than during period 1 mostly because of net precipitation in subdomain B and a net inflow of freshwater in subdomain A.

In my understanding, the analyses that would have answered the question asked by the authors (which is: why the sea level stopped rising during period 2?) would have been to study separately the two periods and their drivers and then compare them. For example, for the whole domain, the positive sea level trend during period one (fig 4a) seems to be due to a positive steric trend (fig 5a) and a positive manometric trend (fig 4b). On the other hand, during period 2, the stabilisation of the SLA (fig 4a) seems to be due to a positive manometric trend (fig 4b) being counterbalanced by a negative steric trend (fig 5a) which is itself due to a negative thermosteric trend (fig 8a). Therefore, we can interpret the pausing in sea level rise during period 2 as a result of a cooling of the domain.

I suggest that the author adjust their narrative so that the results match the scientific question being asked by either modifying the scientific question or the results presented.

Thank you very much for this inciteful comment, and for providing this clarification. As suggested, we have modified the questions asked and the entire manuscript accordingly. The revised questions asked in this study is in line 45–51 of the revised manuscript, copied below for convenience:

Our aim is to characterize the evolution of the sea level and its drivers during a period of rising sea level (the rising state, period one: 1996–2004) and a pause in the sea level rise (the hiatus state, period two: 2010–2018). To do this we performed two related tasks. First, to differentiate the relation among the sea level drivers between the rising and hiatus states, we characterize the evolution of the drivers in each state. Second, we characterize the transition of the sea level and its drivers between the two states. In task one we answer two questions: what caused the sea level to continue rising during period one? What caused the sea level rise to pause during period two? In task two we answer the question: what caused the transition of the sea level from a rising state to a hiatus state?

Section 3.1 and Table A1 addresses task 1, while sections 3.2 and 3.3, and Table A2 and A3 addresses task 2.

In light of these revisions, we have modified the manuscript title for clarity and correctness. The new manuscript title is now

"A multidecadal sea level rise and its hiatus in the tropical Atlantic margin off northwest Africa"

My second major comment is about the potential physical linkage between subdomain A salinity and salinity in the western subpolar north atlantic. In my opinion, the manuscript does not provide enough evidence of the existence of the path 1 and path 2 and the westward flow connecting path 2 and 1. It is not clear to me, to what extent the paths 1 and 2 are already known (please provide citation if so) or if they are an original finding of the authors. These paths could be valid but should be presented with more care (as hypotheses or as potential connections) if they are not demonstrated more convincingly. It would be possible to prove a direct link for example by using water parcels lagrangian backwards tracking in time with particles released in the region of interest.

Thank you for the comment. The Azores current, i.e., path 1, is well known (lines 309–318). The potential vorticity (PV) analysis presented in paper, which shows the origin of water mass with the same characteristics, indeed serves the purpose of Lagrangian particle tracking. We have now included a schematic on the PV ratio figure (Fig. 11b) to more clearly show path 1 and 2. We have also included a time lag figure (Fig. 10b) that gives further support to the existence of path 1 and 2. Nonetheless, we have clarified in the abstract as well as in the manuscript that this physical linkage is a hypothesis inferred from our results. Please see the abstract, Fig. 10b, Fig. 11b, and section 4 in the revised manuscript.

Minor comments:

Throughout the whole manuscript:

Usage of the word 'upward'. I found a bit confusing the usage of 'upward' to refer to the positive shift in sea level between the two periods. The instinctive understanding of 'upward' usually refers to something moved higher in depth (it is used as such lines 9, 29, and 30). I recommend replacing it by 'positive' when referring to a shift when values are greater during period 2 than period 1.

Thank you for the suggestion. We have revised the entire manuscript for clarity as suggested.

Usage of word 'shift': This word is overly used and sometimes its meaning is not clear. I suggest replacing it with words like 'change', 'increase', 'decrease' etc. when relevant.

Thank you for the suggestion. We have implemented the suggestion where appropriate.

The word 'hiatus' is used very early in the manuscript but is never really defined. It is only line 45 than the reader guess than it means 'pause in the sea level rise'. I recommend defining it early in the introduction. In addition, if the authors decide to modify the scientific question of the manuscript following my major comment, I recommend to not use the word 'hiatus' at all as it can be easily replaced by 'pausing in sea level rise' or other simple words.

Thank you for the comment and suggestion. We have now defined hiatus in the abstract (line 3 of the revised manuscript) and provided more clarification on its usage in the introduction (lines 45–46 in the revised manuscript).

All figures showing time series: It would greatly help the reader to highlight period 1 and 2 on all time series either by lightly color-shading the background or delimiting them using vertical lines.

Thank you for the suggestion. We have now revised all the figures to have a light gray-shading background that delineates period 1 and period 2.

line 5: The steric and mass contributions do not sum up to 100% which can be confusing as the reader usually expects it to do so. The reader later understands why (table A1) but I suggest clarifying this in the abstract.

Thank you for the suggestion. We have modified the abstract accordingly (line 6 of the revised manuscript).

Line 20-22: please add a citation to support this sentence.

Thank you. We have added the reference (line 24 of the revised manuscript.)

Line 30: 'Fig 10'. It seems that the figure number is incorrect. Please verify.

Thank you. We have corrected the figure reference (line 32 of the revised manuscript).

Line 35: 'temperature shifted upward'. Does the author mean 'northward'? or increase in temperature? Please clarify.

Thank you for the comment. We have clarified the sentence (line 37 of the revised manuscript).

Line 81-84: It is not clear to me how the vertical land displacement is not taken into account in the study. Is it simply neglected? Is it included in the ORAS5 reanalysis or not? How does this impact the comparison with the altimetry data and grace data?

Thank you for the comment. ORAS5 does not include vertical land motion (vlm). We have now included vlm in our analysis using tidal gauge data sets (lines 132–140), and we have modified the entire manuscript accordingly.

Typos and misspelling:

First paragraph: spaces are missing in units presentation. This would increase readability.

Thank you, we have made the correction.

Line 83: tide gauge is misspelled 'guage'.

Thank you, we have removed this sentence from the manuscript.

Line 162: 'gauge' river is misspelled.

Thank you, we have made the correction (line 186 of the revised manuscript).

Figure 11: 'and' is duplicated in the caption.

Thank you, we have made the correction. Please see the revised Fig. 11 caption.

Line 379: 'here' to replace by 'there'

Thank you, we have made the correction.

Methods:

Line 102: it is possible to simplify the sentence by removing 'taking the difference between two satellite measurements (i.e.,'.

Thank you, we have revised this sentence (line 106 in the revised manuscript).

Line 111: replace 'bathymetry' by 'seabed' or ' ocean floor'

Thank you, we have done it (line 115 in the revised manuscript).

Line 153: The agreement is good between ORAS5 G and Grace data for most of the time series but not in 2003-2005 and from 2015. It would be nice to have some comments on this especially because these years are part of periods one and two.

Thank you for the suggestion. We have done it (lines 175–178 in the revised manuscript), copied below for convenience.

The agreement between ORAS5 G and GRACE datasets is good (Fig. 2[b]). However, there are discrepancies, particularly during 2003–2005 and from 2015, which may be explained by degraded GRACE data when the satellite orbit is near exact repeat (July–Dec 2004, Jan–Feb 2015) and when altimetry measurements are taken from only one accelerometer, starting around Aug 2016 (NASA, Jet Propulsion Laboratory, 2019).

Figure 3: panels a1 and a2 are not referred nor commented on in the text. What is the relevance of these panels?

Thank you for the comment. Figure 3: panels a1 and a2 are the spatial pattern and time series of EOF mode 1 without the annual cycle not removed, which enable reconstruction of the whole pattern of variability during 1985–2017: the temporal pattern shows an annual variability and a long-term steadily rising SLA (lines 191–194 in the revised manuscript).

Results:

Line 202-203: Can you elaborate more to explain why the interpretation is consistent with the SLA pattern in the EOF analysis? This is not evident to me.

Thank you for the comment, and we regret the lack of clarity in the sentence. We meant that the dominance of oceanic processes in subdomain A and dominance of atmosphere-ocean processes in subdomain B is consistent with the differing spatial pattern of SLA variability in subdomain A and subdomain B that is revealed in the EOF analysis (Fig. 3[b1]), lines 247–252 in the revised manuscript.

Lines 208-220: These paragraphs were a bit difficult to read. I suggest reorganizing them to separate the interpretation for domain A and for domain B.

Thank you for the comment, we have revised the paragraphs accordingly. Please see lines 257–270 in the revised manuscript.

Figure 7: I am wondering why analysing the temperature and salinity instead of analysing directly the thermosteric and halosteric sea level. Is it because it is not possible to compute the thermosteric and halosteric sea level for the four layers?

Thank you for the comment. We did this, as a first step, because the temperature and salinity time series shows more clearly the shifts in the vertical structure, which then lead us to the analysis of the steric changes in the study domain.

Figure 9 and 12: It would help the reader to place the domain, subdomain A, and subdomain B on these maps when they are part of the plot to help to locate them. On figure 9 a1, it would also help to have a schematic arrow to represent the guinea done cyclonic circulation.

Thank you for the suggestion, we have done it. Please see the revised Fig. 9 and Fig. 12

Line 273-274: It is said that layers 2 and 3 have 'comparatively large salinity shifts'. I don't understand. It is compared to what? To period 1? To the other layers?

**Thank you for the comment. We regret the lack of clarity in the sentence, and we have rephrased it accordingly (lines 323–326), copied below.**

We chose these layers because, first, layer 2 and 3 account for 71.6% of the total steric expansion in subdomain A (not shown); second, layer 2 and 3 have large halosteric shifts in subdomain A during period two compared to layer 1 (only 50 m thick) and layer 4 (Fig. 7[a2]); and third, short-term air-sea fluxes that modify salinity are mostly attenuated in layers 2 and 3.

Figure 10: panel a. It would be useful to show the time lag corresponding to the maximum correlation shown on panel a. This would strengthen the hypothesis of the potential pathways to link the subpolar North Atlantic and domain A.

Thank you for the suggestion, we have included the time lag figure, Fig. 10b in the revised manuscript.

Line 281-282: the currents located around 45N that would link path 2 to path 1 is in my opinion not visible from fig 11a.

Thank you for the comment, we have replotted the figure in the revised manuscript, Fig. 11.a.

Figure 10b: I am wondering about the relevance of this map of regression coefficients. Are the regression coefficients computed with a time lag (I understand it is not but please clarify)? I also noticed that it is barely used in the manuscript. To me the paths drawn on top of the map are not hinted at by the map itself. In other words, in my opinion, this map does not add any evidence to the possible existence of the paths indicated by the arrows. In addition, from a methodological point of view, it is not clear to me why the linear regression during period 2 between salinity averaged in domain A and the other grid points in the Atlantic would prove a causal link between the two, especially because (as it is said in the manuscript) it would take some time for the salinity in one place to be reflected in the other.

The reviewer is correct, thank you for the suggestion We have removed the regression coefficients figure, and replaced it with the time lag figure.

Line 287-288: Could you provide a justification to why the purple box chosen in the western subpolar north Atlantic is representative?

Thank you for the comment. The purple box is representative, first because it has a high correlation with subdomain A salinity as shown in Fig. 10a. Second, we calculated the mean difference between salinity during 2012–2016 (when a large freshening was reported in the subpolar region) and during 1985–2018, and the purple region has the largest salinity shift in the subpolar region. Please see the attached figure.

[Figure]

FIG. C1. Surface to 200 m average salinity change in the North Atlantic Ocean: mean 2012–2016 salinity minus mean 1985–2018 salinity

Line 291 and following: I am not particularly knowledgeable on potential vorticity. Is the conservation of potential vorticity a common way to try to trace water masses? If so please provide citations.

Thank you for the comment. Yes, potential vorticity is the way to trace the origin of water masses having the same characteristics. Please see the following reference.

Apel, J.: Principles of Ocean Physics, ISSN, Elsevier Science, ISBN 9780080570747, 1987, pages 270–282.

Line 303-305: To me the paths 1 and 2 are not visually clear in figure 11b.

Thank you for the comment. We have added schematic lines to Fig 11b to make path 1 and path2 clearer, which shows the movement of water masses with the same characteristics.

Line 307-308: How is this lagged correlation computed? Is it from the time series of figure 7 a3? In that case it is natural that this is consistent with figure 7 a3.

Thank you for the comment. Yes, it is indeed calculated this way.

Section 5: It seems that for the demonstration of this section the vertical velocities are very important but they are not analysed. Why not also analyse the vertical velocities?

Thank you for the comment. We have now analyzed the vertical velocity in included the time series at the Guinea Dome core, Fig. 13b.

Line 361-364: the shift of the northern edge of the countercurrent is not clear to me. Maybe it would be useful to indicate on the figure.

Thank you for the suggestion. We have indicated the North Equatorial Countercurrent on Fig. 9

Line 364: 'vertical velocity'. The shift of the vertical velocities is not clearly demonstrated but inferred. Please correct the wording.

We have done so. Please also see Fig. 13b

Concluding section:

Point 5: Can you elaborate more on this point? Do you have examples of work that has been done using numerical simulations in that region or elsewhere? Can you hint at what model would be a good option to do this?

Thank you for the comment. We envisage a high-resolution atmosphere and coastal ocean two-way, coupled model to incorporate all current knowledge in this ocean margin, to provide a system for performing numerical experiments that will increase our understanding of the operating mechanism in this vitally important tropical region, and to facilitate better predictions. This type of 3D atmospheric and oceanic coupled model (Ibrahim et al., 2020), when nested with a larger domain ocean model and forced with land-surface hydrological inputs, can be used for experiments to identify and quantify critical processes that are involved in coastal deep ocean exchanges (lines 466–473 in the revised manuscript).

---

## Author Comment (AC2)

**Response to Reviewer 2**

Thank you for reviewing our manuscript and for the inciteful comments and suggestions. We have addressed these comments, as shown below, and with line numbers for reference to the revised manuscript. Our responses are in blue font below each comment.

This study provides a valuable analysis of the 2010–2018 sea level rise hiatus off northwest Africa using satellite/reanalysis data (1993–2018). The identification of steric dominance (74%) and subpolar-tropical salinity linkages is novel and impactful. Methodological rigor and validation with independent datasets are key strengths. However, clarification of physical mechanisms and data limitations is needed for full acceptance. I recommend acceptance after minor revisions.

Thank you very much for your comments and suggestions.

Eq. 6 oversimplifies vertical-horizontal coupling. Could you show decadal trends of vertical velocity (w) in the dome core using ORAS5 data?

Thank you. Yes, we have done so. We have now included the time series of vertical velocity in the Guinea Dome core in the revised manuscript, Fig. 13b.

Please discuss how anticyclonic circulation modulates downwelling against background upwelling in more detail.

The anticyclonic shift will weaken upwelling of cooler water from beneath, resulting in warming in upper layers, and it will cause a convergence zone, resulting in downwelling of surface water that will increase warming beneath, as demonstrated by the shift in seasonal temperature at 50 m during period two (Fig. 12[a3,b3]) and by the shift in temperature and velocity time series at the Guinea Dome core (Fig. 13[a,b]). The downwelling, moreover, is opposite to the permanent upward flux of water in the Guinea Dome, which implies a weakening of the dome circulation between periods one and two.

While mentioned in Section 2.2, error quantification is missing. Could the authors add error bars in Fig. 4?

Thank you for the comment. We have now included in the revised manuscript the uncertainty of the satellite altimetry and GRACE mass data in Fig. 2a and 2b, respectively, as well as the uncertainty of ERA5 P-E and land runoff in Figs. 4c and 4d, respectively.

The 2010–2018 hiatus (Fig. 2a) coincides with subdomain A freshening (Fig. 7a2) but the causal sequencing is unclear. Please add lead-lag correlation analysis between subpolar salinity and SLA.

Thank you for the comment. Because of mixing during the propagation of subpolar waters to subdomain A, and because subdomain A *SLA* integrates other factors aside from salinity shifts, the physical basis for correlating subpolar salinity and subdomain A. *SLA* is somewhat weak: it is difficult for the propagating water to retain its characteristics during transport; for example, the temperature changes in the surface layer due to the radiation, which could modulate subdomain A SLA. Instead, we have correlated subpolar salinity and subdomain A salinity in the same layer (Fig. 7a3), which gives a coefficient of 0.687 when subdomain A salinity is lagging by about 5.5 years (lines 362–364). The physical justification for doing this is that, after excluding the probability of all local sources of water, it is only water from a remote source that can modulate subdomain A salinity.